# Later life outcomes of women by adolescent birth history: analysis of the 2016 Uganda Demographic and Health Survey

Dinah Amongin [1,2] Anna Kågesten,[3] Özge Tunçalp,[4] A Nakimuli,[2] Mary Nakafeero,[5] Lynn Atuyambe,[6] Claudia Hanson,[3,7] Lenka Benova[8,9]

For numbered affiliations see end of article.

**Correspondence to**
Dr Dinah Amongin;
amongdinah2003@yahoo.com

## ABSTRACT

**Objectives** To describe the long-term socioeconomic and reproductive health outcomes of women in Uganda by adolescent birth history.

**Design** Cross-sectional study.

**Setting** Uganda.

**Participants** Women aged 40–49 years at the 2016 Uganda Demographic and Health Survey.

**Outcome measures** We compared socioeconomic and reproductive outcomes among those with first birth <18 years versus not. Among those with a first birth <18 years, we compared those with and without repeat adolescent births (another birth <20 years). We used two-sample test for proportions, linear regression and Poisson regression.

**Findings** Among the 2814 women aged 40–49 years analysed, 36.2% reported a first birth <18 years and 85.9% of these had a repeat adolescent birth. Compared with women with no birth <18 years, those with first birth <18 years were less likely to have completed primary education (16.3% vs 32.2%, p<0.001), more likely to be illiterate (55.0% vs 44.0%, p<0.001), to report challenges seeking healthcare (67.6% vs 61.8%, p=0.002) and had higher mean number of births by age 40 years (6.6 vs 5.3, p<0.001). Among women married at time of survey, those with birth <18 years had older husbands (p<0.001) who also had lower educational attainment (p<0.001). Educational attainment, household wealth score, total number of births and under-5 mortality among women with one adolescent birth were similar, and sometimes better, than among those with no birth <18 years.

**Conclusions** Results suggest lifelong adverse socioeconomic and reproductive outcomes among women with adolescent birth, primarily in the category with repeat adolescent birth. While our results might be birth-cohort specific, they underscore the need to support adolescent mothers to have the same possibilities to develop their potentials, by supporting school continuation and prevention of further unwanted pregnancies.

## Strengths and limitations of this study

► We used a nationally representative sample of women in Uganda and assessed the extent of differential recall in reporting of adolescent births (which was minimal).

► The cross-sectional nature of the data and our descriptive analysis did not allow for assessment of causality.

► The results may largely apply to this cohort due to societal changes and advancements in protection of women but, this is a critical starting point to examine long-term impact of early birth and repeat adolescent birth.

of sub-Saharan Africa, the levels of adolescent childbearing in Uganda remain unacceptably high: 24.8% of women aged 15–19 years have begun childbearing (pregnant or have given birth), levels similar to 15 years ago.[6–9] According to the 2016 Uganda Demographic and Health Survey (DHS), 28.4% of women aged 20–24 years had an adolescent birth before age 18 years, a decline from 41.7% in 1988/1989. Among women aged 20–24 years for whom the first birth occurred before age 18 years, the percentage reporting repeat adolescent birth (first birth <18 years, and another birth before the age of 20 years) did not decline in 30 years (58.9% in 1988/1989, 55.6% in 2016).[10]

Adolescence (ages 10–19 years) is characterised by rapid physical, social, cognitive and emotional development that lays the foundation for an individual's health and well-being across the life course. As noted by Patton *et al*,[11] adolescence is a period when individuals acquire health, social, cultural, financial and educational assets, whose effects can spill over to the next generation.[11]

Global evidence shows that childbirth in adolescence, especially before the age of 18

## INTRODUCTION

The sub-Saharan Africa region continues to have the world's highest burden of adolescent births, accounting for 95% of the 12 million births occurring to women aged 15–19 years old each year globally.[1–5] Just as in other parts

years, is associated with poor social, economic and health outcomes for both the girl and her offspring.[12–15] The health challenges associated with adolescent birth include obstructed labour, postpartum haemorrhage, preterm birth, fistulae, sepsis and infant death among others.[5 12 16] In addition, adolescent childbirth can negatively affect socioeconomic outcomes leading to school dropout, limited cash income and forced/early marriage.[17–20] Married adolescents are more likely than those unmarried to initiate childbearing early, permanently drop out of school, and lack autonomy over their sexuality and reproduction due to patriarchal control.[21–23] Younger adolescent girls are more prone to the negative outcomes due to their immature physiological state, poor economic independence and developing cognition.[24 25] Having repeat pregnancies during adolescence can further exacerbate the risk of such negative outcomes.[26 27]

While the immediate effects are relatively described, there is limited evidence of the long-term socioeconomic and reproductive health outcomes following early initiation of childbearing, including the effects of repeat adolescent birth, in Uganda and other low-income countries.[28 29] The available information, mainly from high-income countries, suggests lifelong persistence of socioeconomic disadvantage such as low educational attainment, poor earnings and less stable marriages following an adolescent birth.[30–33] A Population Council and Women Deliver report explored the short-term and long-term socioeconomic (employment and cash earnings) impact of having a child <18 years using the most recent DHS data sets from 43 low/middle-income countries (LMICs) including the 2016 Uganda DHS.[33] This report suggested that economic disadvantages of adolescent childbirth persist over the entire lifetime for women. A study in Mexico found that women aged 25–64 years who had a birth between ages 15 and 19 years had lower educational attainment and lower income compared with those who did not.[32] Global literature points to persistence of the negative effects of adolescent birth on a woman's social welfare and household, including into the next generations.[2 5 12 34] A woman's agency and decision-making power can be negatively affected. For example, the resultant low educational and wealth attainment following early birth[5] is associated with high unmet need for contraception.[35] In the long term, women who start childbirth during adolescence tend to end up with older partners who have lower educational and socioeconomic attainment.[31 36 37] However, these studies did not disaggregate outcomes among women who had *repeat* adolescent birth following the first birth <18 years, nor did they assess outcomes related to women's lifelong reproductive health.

In response to this gap, this paper seeks to investigate the socioeconomic and reproductive outcomes among women in Uganda toward the end of reproductive life course (40–49 years) according to their adolescent birth history (no birth <18 years, first birth <18 years, repeat births <20 years). We hypothesise that having an adolescent

birth before the age of 18 years leads to lifelong disruption in the acquisition of socioeconomic resources and well-being. Further, we hypothesise that repeat adolescent births (first birth before the age of 18 years and second before the age of 20 years) further worsen these outcomes. Given the link between early marriage and childbirth and its potential influence on later life outcomes,[31 38] we also stratified outcomes by marital status among women with first birth <18 years (comparing those with and without repeat adolescent birth).

## METHODS
### Data sources and population
We used data from the 2016 Uganda DHS individual women's data set. The DHS are nationally representative cross-sectional surveys where multistage cluster sampling is conducted. All the geographical regions of the country were represented. The DHS are conducted every 5 years in Uganda and collect information on population health along with socioeconomic and demographic characteristics. All women aged 15–49 years in sampled households for the individual women's data set provided self-reported information about their live births. The interviewer-administered questionnaires used were translated into local languages and pretested prior to data collection.

The study population included women aged 40–49 years at the time of the 2016 survey, whom we categorised into three adolescent birth histories: (1) no birth <18 years; (2) first birth <18 years and no additional births <20 years; and (3) first birth <18 years and one or more additional births <20 years (repeat adolescent birth). We included live births rather than pregnancies and analysed all women, including those who never gave birth, given that some pregnancies may end in miscarriage/abortion and others in stillbirths and therefore, the later life outcomes may not be impacted by this. To validate women's recall of adolescent childbearing, we conducted a sensitivity analysis comparing the childbirth patterns of women born in 1967–1976 as reported in the 1995 Uganda DHS to the same birth cohort of women as reported in 2016 (online supplemental table 1). We found similar estimates of first birth <18 years and repeat and no repeat adolescent births, indicating similar estimates of adolescent birth history among two representative samples of women from the same birth cohort in 1995 vs 2016.

### Measures
#### Outcome variables
We assessed two main categories of later life outcomes: socioeconomic and reproductive health related. Online supplemental table 2 contains the different variables assessed and their definitions. Socioeconomic outcomes included: women's educational attainment (education level, mean years of education); literacy; household wealth score (calculated using the household assets and presented as a linear index)[39] and receiving cash income at time of the 2016 Uganda DHS. Reproductive health

outcomes included: number of live births by exact age 40 years; under-5 mortality rate among any of the women's children; unmet need for contraception at the time of survey and report of challenge seeking healthcare.

We further analysed outcomes measured only in the subsample of women who were married or in a union at the time of the 2016 survey, including: age difference with husband; husband's education level (categorised in the same way as for women); and several measures of empowerment based on women's reports of whether they (1) solely decide on how their own earnings are spent, (2) are part of a marriage with other wives (polygamy) and (3) decide about their own healthcare.

Background variables included: women's area of residence (urban or rural); geographical region (Central, Eastern, Northern and Western); religion (Anglican, Catholic, Muslim, Other); marital status at survey (currently married/in union, divorced/separated, never in union); and the mean age (years) at survey, first sex, first birth, and at first union (if ever married/in union).

### Statistical analysis

We began with exploratory data analysis to assess the distribution in outcomes and background characteristics and examine missingness. Next, we calculated the percentage of women according to their adolescent birth history (with vs without first birth <18 years, and with vs without repeat adolescent birth). Column percentages of the later life outcomes and their 95% CIs for each category of adolescent birth history were presented. We used the two-sample test of proportions to test for differences in the two column proportions for each later life outcome, and linear regression to compare group means for continuous variables. We further used Poisson regression to compare under-5 mortality rates, estimating incidence risk ratios (IRRs) and associated p values. We described later life outcomes by adolescent birth history among all women and among the subsample of women who were married/in a union at time of survey (for whom partner information was available). We further described later life outcomes in a subanalysis among women with and without repeat adolescent birth by marital status at first birth. For this subanalysis, we included variables with p value of <0.05 between women with and without repeat adolescent births. Information on marital status at first birth was missing for N=95 (N=10 with no repeat adolescent birth and N=85 with repeat adolescent birth); these observations were therefore excluded from the subanalysis.

We used survey weights, stratification and clustering to adjust for the complex survey design and non-response. All analyses were conducted using STATA V.12.0 (StataCorp, Texas, USA).

### Patient and public involvement

For this study, we used secondary data and there was therefore no direct patient or public involvement. However, the study objectives and design were developed in collaboration with the Reproductive Health Division of the Ugandan Ministry of Health for whom adolescent fertility is a key research priority. The study builds on a previous landscape analysis of adolescent health needs conducted by the Ministry of Health, which indicated information gaps related to the long-term effects of adolescent childbearing, especially for repeat adolescent birth. The results from the study will be disseminated to different stakeholders at the regional and national level.

### RESULTS

From the analytical sample of 2814 women aged 40–49 years in the 2016 Uganda DHS, 36.2% had their first birth <18 years (figure 1). Among these, 85.9% reported a repeat adolescent birth (31.1% of all women). Similar proportions were observed for women who were married at the time of survey: 36.4% reported a first birth <18 years and of these, 87.1% reported a repeat adolescent birth (31.7% of all married women).

Table 1 shows the background characteristics of women in the 2016 sample based on their adolescent birth history. Most resided in rural areas (79.1%), belonged to the two main Christian religious faiths (Catholic and Anglican, 86.6%), and had ever been married or in a union (98%). The background characteristics were similar for women with and without first birth <18 years in terms of region, residence, religion, marital status and mean age at survey. However, the mean age at first sex, first birth and first union differed by adolescent birth history. Women with first birth <age 18 years reported earlier age at sexual debut (14.4 years vs 17.3 years), and had their first birth and union earlier (15.5 and 16.3 years) compared with those who began childbearing after age 18 years. In contrast, women with repeat adolescent birth entered into marriage/union around the same age (15.9 years) as their first birth (15.5 years), whereas for those without repeat adolescent birth, first marriage or union (18.3 years) happened on average 2 years after their first birth (16.1 years). The average age at first birth for all women was 19.0 years whereas it was 15.5 years among those with first birth <18 years.

### Later life outcomes by adolescent birth history

Looking at women, all aged 40–49 years (N=2814), results from bivariate analysis indicated that those who had their first birth <18 years had significantly lower mean years of schooling (3.6 vs 5.0 years, p<0.001), were less likely to only have completed primary school or higher (16.3% vs 32.2%, p<0.001), and more likely to be illiterate (55.0% vs 44.0%, p<0.001) compared with those without a birth <18 years (table 2). Further, they also had lower household wealth score (p=0.043), were more likely to report healthcare-seeking challenges (p=0.002) and had a higher mean number of live births by exact age 40 years (6.6 vs 5.3, p<0.001). There were no differences between the two groups in children's under-5 mortality rate.

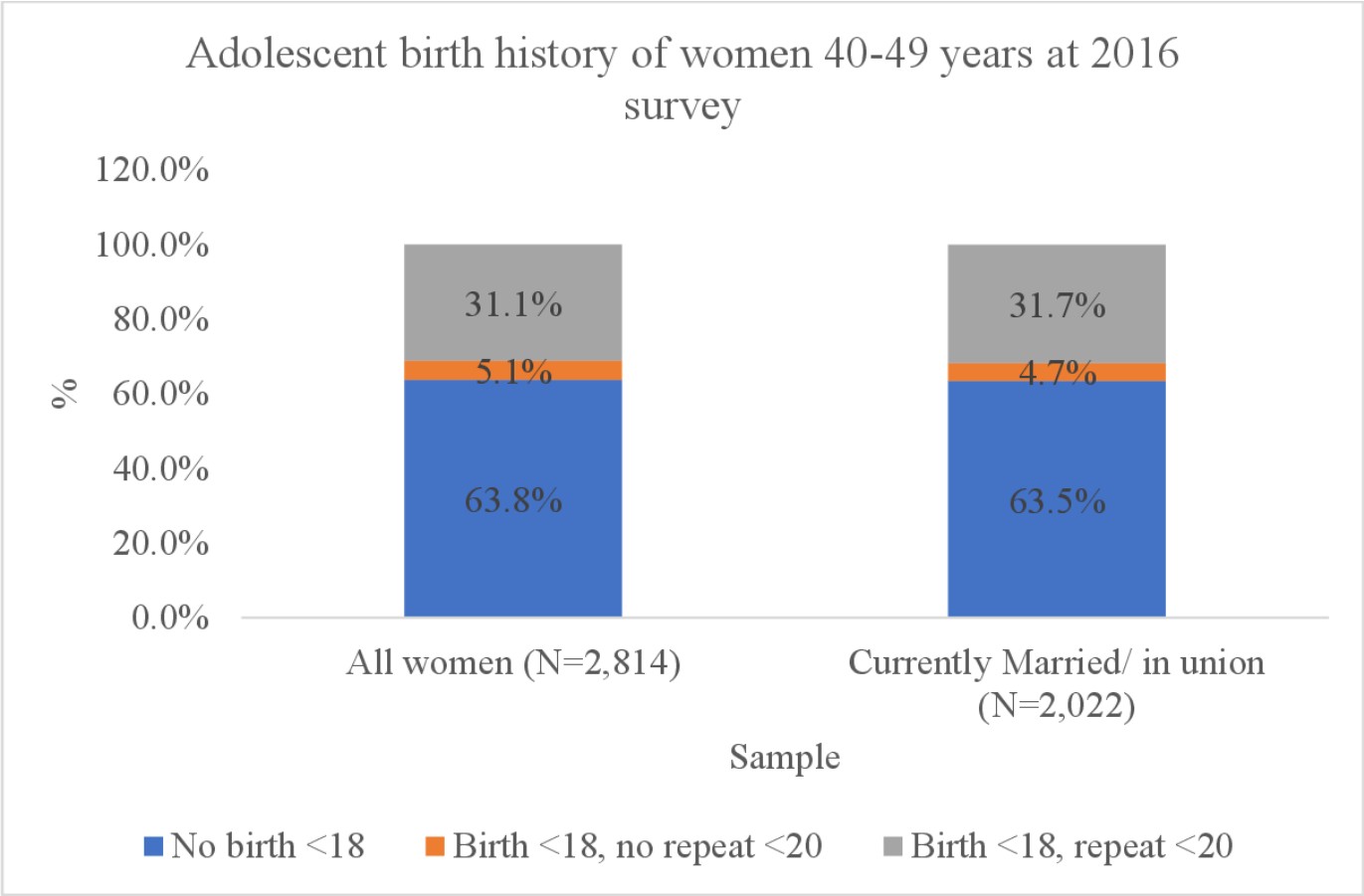

**Figure 1** Adolescent birth history among Ugandan women aged 40–49 years at the 2016 Uganda DHS. DHS, Demographic and Health Survey.

Among the 1020 women with first birth <18 years, those with repeat adolescent birth were less likely than those without such experiences to have completed at least primary education (14.2% vs 28.9%, p<0.001), to have lower mean years of education (3.4 vs 4.7 years, p<0.001), household wealth scores (−0.003 vs 0.020, p=0.032), more likely to be illiterate (57.1% vs 41.9%, p<0.001) and to have higher mean number of live births by age 40 years (7.0 vs 4.0, p<0.001). Further, the under-5 mortality rate for children born to women with repeat adolescent births was higher than for those born to women with no repeat adolescent birth (IRR=1.36, 95% CI 1.06 to 1.75). Healthcare-seeking challenges and unmet need for contraceptives were, however, similar when comparing women with and without repeat adolescent births.

### Analysis of a subsample of women 40–49 years married at time of 2016 survey

We further conducted analysis among a subsample of women who were married at time of the 2016 survey (N=2022). Compared with women with no birth <18 years, those with first birth <18 years had partners/husbands who were older (7.6 vs 5.4 years older, p<0.001) and with lower education level (54.1% vs 61.2% primary school or above, p=0.002) (table 3). Further, more women with first birth <18 years had control over how their own

earnings are spent (p<0.001). There were no differences in outcomes regarding being in polygamous marriage and non-inclusion in decision-making about women's own healthcare. Among the 737 married women who had a first birth <18 years, there were no other differences in outcomes observed by whether they had repeat adolescent births.

### Analysis of women with and without repeat adolescent births, by marital status at first birth

We also conducted analysis of later life outcomes among women with and without repeat adolescent birth by their marital status at first birth, including variables that were significantly associated with repeat adolescent birth (p<0.05) among all women aged 40–49 years (table 2). The proportion of married at first birth was higher for those with versus without repeat adolescent birth (76.4% vs 63.4%, p=0.001) (table 4). Results showed no differences in later life outcomes based on marital status among women without repeat adolescent births. However, among women with repeat adolescent birth, those who were married at first birth had lower mean years of schooling (3.2 vs 3.9 years, p=0.017) and were more likely to be illiterate (59.7% vs 49.3%, p=0.012) compared with those who were not married at the time.

**Table 1** Background characteristics of Ugandan women 40–49 years by adolescent birth history; 2016 Uganda DHS

| | All women 40–49 years col% (95% CI) | No birth <18 years col% (95% CI) | First birth <18 years col% (95% CI) | Birth <18 years, no repeat col% (95% CI) | Birth <18 years, repeat col% (95% CI) |
|---|---|---|---|---|---|
| **Variables** | N=2814 | N=1795 | N=1020 | N=144 | N=876 |
| **Residence** | | | | | |
| Urban | 20.9 | 21.0 | 20.8 | 23.7 | 20.3 |
| | (18.7 to 23.3) | (18.3 to 23.9) | (17.7 to 24.3) | (16.4 to 33.1) | (17.4 to 23.7) |
| Rural | 79.1 | 79.0 | 79.2 | 76.3 | 79.7 |
| | (76.7 to 81.3) | (76.1 to 81.7) | (75.7 to 82.3) | (66.9 to 83.6) | (76.3 to 82.6) |
| **Region** | | | | | |
| Central | 26.5 | 24.4) | 30.1 | 31.4 | 29.9 |
| | (22.3 to 31.1) | (20.2 to 29.3) | (24.8 to 36.0) | (21.8 to 42.9) | (24.3 to 36.1) |
| Eastern | 30.3 | 29.4 | 31.9 | 30.2 | 32.2 |
| | (26.1 to 34.9) | (25.1 to 34.1) | (26.8 to 37.6) | (21.4 to 40.6) | (26.9 to 38.1) |
| Northern | 17.6 | 18.3 | 16.5 | 18.1 | 16.2 |
| | (14.6 to 21.1) | (15.1 to 22.1) | (13.2 to 20.2) | (12.3 to 26.0) | (12.9 to 20.0) |
| Western | 25.6 | 27.9 | 21.5 | 20.3 | 21.7 |
| | (21.7 to 29.8) | (23.5 to 32.7) | (17.6 to 26.0) | (13.8 to 28.9) | (17.6 to 26.4) |
| **Religion** | | | | | |
| Anglican | 44.6 | 44.8 | 44.1 | 39.4 | 44.8 |
| | (42.0 to 47.2) | (41.9 to 47.9) | (40.1 to 48.1) | (30.6 to 48.9) | (40.6 to 49.2) |
| Catholic | 42.0 | 43.2 | 40.0 | 48.0 | 38.7 |
| | (39.2 to 44.9) | (39.9 to 46.5) | (36.2 to 43.9) | (38.9 to 57.2) | (34.6 to 43.0) |
| Muslim | 10.5 | 9.1 | 13.0 | 9.2 | 13.7 |
| | (8.8 to 12.6) | (7.2 to 11.4) | (10.4 to 16.2) | (5.3 to 15.7) | (10.8 to 17.1) |
| Other | 2.9 | 2.9 | 2.9 | 3.4 | 2.8 |
| | (2.2 to 3.8) | (2.1 to 4.0) | (1.8 to 4.5) | (1.2 to 9.2) | (1.7 to 4.5) |
| **Marital status at survey** | | | | | |
| Never in union | 2.0 | 2.4 | 1.3 | 0.3 | 1.4 |
| | (1.5 to 2.7) | (1.7 to 3.4) | (0.7 to 2.3) | (0.0 to 2.0) | (0.8 to 2.6) |
| Currently in union/marriage | 71.8 | 71.6 | 72.3 | 66.4 | 73.3 |
| | (69.9 to 73.7) | (69.1 to 73.9) | (69.0 to 75.4) | (57.1 to 74.5) | (69.8 to 76.5) |
| Formerly in union/marriage | 26.2 | 26.0 | 26.4 | 33.4 | 25.3 |
| | (24.3 to 28.1) | (23.7 to 28.5) | (23.4 to 29.7) | (25.2 to 42.6) | (22.1 to 28.7) |
| Mean age at survey (SD) | 43.9 (2.9) | 44.1 (2.8) | 43.8 (2.9) | 43.5 (2.9) | 43.9 (2.9) |
| Mean age at first sex (SD) | 16.3 (2.8) | 17.3 (2.8) | 14.4 (1.5) | 15.1 (1.4) | 14.3 (1.5) |
| Mean age at first birth (SD) | 19.0 (3.9) | 21.1 (3.3) | 15.5 (1.5) | 16.1 (1.2) | 15.5 (1.5) |
| Mean age at first union, if ever married (SD) | 18.9 (5.3) | 20.3 (4.9) | 16.3 (5.0) | 18.3 (5.8) | 15.9 (4.8) |

DHS, Demographic and Health Survey.

## DISCUSSION

We investigated the socioeconomic and reproductive health outcomes among Ugandan women towards the end of their reproductive life using data from the most recent Uganda DHS collected in 2016. We compared women's outcomes according to their adolescent birth histories: those with and without first birth <18 years, and among those with first birth <18 years, those with and without repeat adolescent births (two or more births before age 20 years). For the latter groups, we also examined differences in later life outcomes by marital status at the point of the first adolescent birth.

Our findings highlight three key points. First, women who had their first birth <18 years reported more negative

**Table 2** Socioeconomic and health-related life-course outcomes by adolescent birth history among Ugandan women 40–49 years, 2016 Uganda DHS (N=2815)

| | All women 40–49 years | | | Women with first birth <18 years | | |
|---|---|---|---|---|---|---|
| | No birth <18 years (N=1795) | Birth <18 years (N=1020) | | No repeat adolescent birth (N=144) | Repeat adolescent birth (N=876) | |
| Variables | % (95% CI) | % (95% CI) | P value | % (95% CI) | % (95% CI) | P value |
| Education level | | | | | | |
| Complete primary and above | 32.2 (29.3 to 35.3) | 16.3 (13.7 to 19.2) | <0.001 | 28.9 (21.2 to 37.9) | 14.2 (11.6 to 17.3) | <0.001 |
| Mean years in school (SD) | 5.0 (4.4) | 3.6 (3.2) | <0.001 | 4.7 (3.4) | 3.4 (3.1) | 0.001 |
| Literacy | | | | | | |
| Illiterate (cannot read at all) | 44.0 (40.9 to 47.1) | 55.0 (50.8 to 59.1) | <0.001 | 41.9 (32.7 to 51.8) | 57.1 (52.8 to 61.4) | <0.001 |
| Household wealth score | 0.009 (0.100) | 0.001 (0.085) | 0.043 | 0.020 (0.099) | −0.003 (0.082) | 0.032 |
| Receiving cash income at survey (yes) | 70.0 (67.0 to 72.8) | 70.5 (67.1 to 73.7) | 0.780 | 71.3 (61.7 to 79.2) | 70.4 (66.7 to 73.8) | 0.826 |
| Challenge seeking healthcare (yes) | 61.8 (58.6 to 64.9) | 67.6 (63.9 to 71.1) | 0.002 | 62.0 (52.2 to 70.9) | 68.5 (64.5 to 72.3) | 0.123 |
| Mean no of live births by exact age 40 years (SD) | 5.3 (2.1) | 6.6 (1.7) | <0.001 | 4.0 (1.6) | 7.0 (1.3) | <0.001 |
| U5 mortality rate | 124.6 | 140.1 | 0.341 | 104.8 | 142.9 | 0.016 |
| Unmet need for contraception | 19.8 (17.8 to 21.9) | 20.2 (17.4 to 23.4) | 0.799 | 20.1 (13.0 to 29.6) | 20.2 (17.1 to 23.8) | 0.978 |

Under-5 mortality: divided by 1000 live births. P value obtained for the IRR that compared the mortality rate between two groups. P=0.016, IRR was 1.36 (95% CI=1.06 to 0.75).
DHS, Demographic and Health Survey; IRR, incidence rate ratio.

socioeconomic and reproductive health outcomes in later life than those without such experiences. These differences appear to be driven by the largest subgroup of women with repeat adolescent birth, who had poorer educational attainment, lower household wealth and empowerment, as well as higher fertility and under-5

**Table 3** Later life outcomes related to partner characteristics among a subsample of married women 40–49 (2016 Uganda DHS, N=2022)

| | All women 40–49 years | | | Women with first birth <18 years | | |
|---|---|---|---|---|---|---|
| | No birth <18 years (N=1285) | Birth <18 years (N=737) | | No repeat adolescent birth (N=096) | Repeat adolescent birth (N=642) | |
| Variables | % (95% CI) | % (95% CI) | P value | % (95% CI) | % (95% CI) | P value |
| Mean age difference with partner/husband-years (SD) | 5.4 (6.4) | 7.6 (7.2) | <0.001 | 6.8 (7.1) | 7.7 (7.2) | 0.358 |
| Husband's education level | N=1285 | N=737 | | N=96 | N=642 | |
| Primary | 54.1 (50.5 to 57.7) | 61.2 (56.9 to 65.3) | 0.002 | 60.0 (48.6 to 70.5) | 61.3 (56.8 to 65.6) | 0.808 |
| Secondary/higher | 34.1 (30.7 to 37.7) | 26.6 (22.9 to 30.6) | <0.001 | 31.7 (22.2 to 43.0) | 25.8 (22.0 to 30.0) | 0.222 |
| Women who solely decide on how her earnings are spent | N=897 | N=499 | <0.001 | N=68 | N=499 | 0.910 |
| | 52.1 (48.3 to 55.9) | 64.2 (58.8 to 69.3) | | 64.8 (50.2 to 77.0) | 64.1 (58.6 to 69.3) | |
| Women not included on decisions about: | | | | | | |
| Their own healthcare | 18.8 (16.4 to 21.4) | 20.9 (17.5 to 24.8) | 0.252 | 15.5 (8.7 to 26.0) | 21.8 (17.9 to 26.2) | 0.157 |
| In marriage with other wives (polygamy) | 33.1 (29.8 to 36.6) | 36.2 (32.3 to 40.2) | 0.157 | 42.9 (31.1 to 55.6) | 35.2 (31.1 to 39.4) | 0.143 |

DHS, Demographic and Health Survey.

**Table 4** Later life outcomes among women 40–49 years with and without repeat adolescent birth, by marital status at first birth (N=925)

| | No repeat adolescent birth | | | Repeat adolescent birth | | |
|---|---|---|---|---|---|---|
| | Not married at first birth (N=49) | Married at first birth (N=85) | | Not married at first birth (N=186) | Married at first birth (N=604) | |
| Variables | % (95% CI) | % (95% CI) | P value | % (95% CI) | % (95% CI) | P value |
| **Education level** | | | | | | |
| Complete primary and above | 29.6 (17.5 to 45.4) | 24.7 (15.7 to 36.5) | 0.536 | 17.4 (12.1 to 24.4) | 12.4 (9.7 to 15.8) | 0.082 |
| Mean years in school (SD) | 4.7 (3.0) | 4.2 (3.2) | 0.458 | 3.9 (3.0) | 3.2 (3.1) | 0.017 |
| Literacy | | | | | | |
| Cannot read at all (illiterate) | 40.3 (25.2 to 57.5) | 44.4 (32.2 to 57.4) | 0.644 | 49.3 (40.1 to 58.6) | 59.7 (54.7 to 64.5) | 0.012 |
| Household wealth scores (SD) | 0.005 (0.082) | 0.024 (0.097) | 0.320 | 0.004 (0.085) | −0.008 (0.078) | 0.097 |
| Mean no of live births by exact age 40 years (39.99) (SD) | 4.0 (1.7) | 4.2 (1.6) | 0.602 | 7.0 (1.2) | 7.0 (1.3) | 0.963 |
| U5 mortality rate/1000 live births | 109.3 | 107.1 | 0.884 | 140.4 | 144.6 | 0.803 |

P value of the IRR obtained from comparing the mortality rates. P=0.884 (IRR 0.98, 95% CI=0.75 to 1.28) and p=0.803 (IRR 1.03, 95% CI=0.82 to 1.30).
IRR, incidence rate ratio.

infant mortality compared with the other groups. This finding aligns with previous research highlighting adolescent childbirth as both a risk factor for, and consequence of, poverty and low education.[5 40 41] School discontinuation due to pregnancy is a persistent problem,[41] and once a girl has given birth, she needs social, psychosocial and economic support to go back to school and complete her secondary education.[5] Our findings further suggest that the risk of school discontinuation was greatest for girls who were married at first birth and went on to have repeat adolescent births, compared with their unmarried counterparts. It is possible that married adolescents with repeat adolescent birth are taken out of school entirely,[33 41] underscoring the need to prevent early marriage in a country where marriage before age 18 years remains common (40%),[42 43] and thus a risk factor for repeat adolescent birth and its negative later life outcomes.

Second, women with an adolescent birth, and especially with repeat adolescent birth, were more likely to be socioeconomically disadvantaged in later life—confirming the limited evidence on poor economic trajectories following early childbearing.[32] For example, the Population Council and Women Deliver analysis of data from 43 LMICs found that adolescents who gave birth before age 18 years were significantly less likely to be earning cash, although there were no differences in employment.[44] This suggests that low educational attainment and the burden of caring for many children ultimately lowered women's agency and chances of economic empowerment. This position is further supported by our finding of similar, or even better, socioeconomic outcomes among women with

one adolescent birth compared with those with no birth <18 years; the lower mean number of children provided avenues for these women to pursue economic activities. Having a partner with a low educational attainment, as was the case for those with repeat adolescent births, may further have worsened the negative economic cycle. Indeed, women's opportunity for economic empowerment is not only linked to their educational attainment and literacy, but to the status of the spouse, which in turn impacts household wealth scores.[33 45 46] Our findings thus confirm the well-demonstrated link between poverty, low education and high fertility globally,[5] calling for renewed efforts to alleviate household poverty and maintain adolescent girls in school.

We also found that women with first birth <18 years were more likely to report greater decision-making power over their own earnings, underscoring the results from the Population Council and Women Deliver report which found that adolescent birth was associated with more control over own economic assets. This suggests that women with an adolescent birth might learn to be self-reliant and independent, providing a potential avenue for interventions to economically empower these women.

Third, our study results suggest that having a large completed family size is driven not only by an early initiation of childbearing, but by repeat adolescent births. Although the adolescent fertility rate has declined in Uganda, it remains high at 132 births per 1000 women aged 15–19 years.[9] This is reflected in how cultural norms in Uganda promote early marriage and childbearing, despite the legislation against early marriage.[38 41 47 48] The household poverty makes it worse, as girls are forced

to discontinue schooling and get sent off into union in exchange for bride price.[43 49 50] Low access and high unmet need for contraceptives among adolescent girls and young women in Uganda further restrict girls' reproductive agency in a heavily patriarchal context.[51] Even in situations where an adolescent girl may want to use contraception, the partner or health worker may object to allow her to access the method.[52] Further, abortion in Uganda is restricted to particular medical conditions, forcing girls facing an unintended pregnancy to either give birth or risk unsafe abortion.[53 54]

The care of a child born to an adolescent girl takes broadly two patterns in Uganda and this cultural position needs to be factored in when viewing outcomes of women with early birth: those in and out of union. When a woman gives birth within marriage, the responsibility of raising the child is usually taken on by the man or his family with the support of the woman.[55] In contrast, when they conceive outside union, the woman and child are usually sent off by the family to live with the man responsible for the pregnancy.[17 56 57] In this situation, the man is assumed to take on the responsibility of raising the child. These cultural norms have remained essentially unchanged over the last four decades[49] despite advocacy for more paternal involvement in the care of children, most especially in scenarios of no union/marriage. Women out of union would most probably have been left with the entire burden of the offspring(s).

### Suggestions for policy change and future research

Public health interventions are needed to prevent repeat adolescent birth and potential negative life-course outcomes via improved access to continued schooling and higher education, and effective contraceptive services. Development of a Uganda school continuation policy for adolescent mothers needs to be fast tracked followed by: wide dissemination, sensitisation of the communities and parents, and enforcement at school level.[48–51 58] Contraception information and services need to be made more accessible to all adolescents in a manner that promotes their autonomy. Further, adolescent women should be protected from early marriage by keeping them in school and strengthening the legislation against early marriage.[47 49 59] Our findings call for more in-depth research to explore and define what women without repeat adolescent birth did that enabled them to recuperate from the effects of early initiation of birth. Further, qualitative studies are needed to explore motivators and circumstances for repeat adolescent birth in sub-Saharan Africa. Finally, we suggest that secondary level completion among adolescent mothers be investigated further among younger cohorts, especially now that the Ministry of Education in Uganda, and many other countries in sub-Sahara Africa, permit girls to continue/resume schooling following pregnancy and childbirth.[40]

### Strength and limitations

This study used data from a large nationally representative sample containing a variety of variables. However, the cross-sectional data cannot support causal associations between adolescent childbirth and the outcomes examined. While our assessment of differential reporting of adolescent childbirth did not show variation on a population level, the results may be influenced by reporting and recall bias due to the self-reported data. Women may also have given socially desirable responses regarding the age at first birth.[60] In addition, some risk factors may have been present right from early childhood. Due to this reverse causality, we cannot be certain of what came first: the negative outcomes or the birth. Nonetheless, this nationally representative survey data are a starting point to investigate later life outcomes in the absence of a prospective cohort in LMIC setting. Additionally, circumstances and opportunities available to women approximately 20 years ago may have altered and therefore, it may not fully depict the current situation in Uganda.

### CONCLUSION

Findings from this study, while not necessarily causal, suggest that among women in Uganda, giving birth before age 18 years is linked to lower educational attainment, household wealth, and higher fertility and under-5 infant mortality outcomes later in life. Further, they tend to have spouses who are older and of low socioeconomic attainment. This pattern is especially strong for women who have repeat adolescent births, and particularly for those who were married at first birth. Women who begin childbearing before age 18 years, but do not proceed to have a repeat adolescent birth—despite being a minority (5%)—appear to have recovered from any negative effects, and in some instances had better outcomes than those with no birth <18 years. This suggests persistence of the negative outcomes thereby underscoring the need to not only prevent early adolescent birth but prevent repeat adolescent births.

**Author affiliations**
$^1$Department of Health Policy Planning and Management, Makerere University School of Public Health, Kampala, Uganda
$^2$Department of Obstetrics and Gynaecology, School of Medicine, Makerere University College of Health Sciences, Kampala, Uganda
$^3$Department of Global Public Health, Karolinska Institute, Solna, Sweden
$^4$Department of Sexual and Reproductive Health and Research, UNDP/UNFPA/UNICEF/WHO/World Bank Special Programme of Research, Development and Research Training in Human Reproduction (HRP), WHO, Geneva, Switzerland
$^5$Department of Epidemiology and Biostatistics, Makerere University School of Public Health, Kampala, Uganda
$^6$Department of Community Health and Behavioral Sciences, Makerere University School of Public Health, Kampala, Uganda
$^7$Department of Disease Control, London School of Hygiene and Tropical Medicine, London, UK
$^8$Department of Public Health, Institute of Tropical Medicine, Antwerpen, Belgium
$^9$Faculty of Epidemiology and Population Health, London School of Hygiene and Tropical Medicine, London, UK

**Contributors** DA conceived and designed the study, obtained permission to use data set from DHS program, conducted analysis, presented and interpreted results, drafted and finalised the article. AK conceived the study, conducted the literature review, reviewed and interpreted the results, and reviewed the article. OT conceived the study, drafted the analysis plan, reviewed and interpreted the results, and reviewed the article. AN was involved reviewing the study design, the results and drafting the article. MN was involved in data analysis, presentation and interpretation of the results. LA was involved in reviewing and interpreting the results, and reviewing the manuscript. CH was involved in designing the study and the analysis, presenting and interpreting the results, drafting the article and reviewing the manuscript. LB was involved in conceiving the study, designing the analysis plan and data analysis, interpreting the results and substantially reviewing the article. All the authors reviewed and approved the manuscript. All the authors take responsibility for their contributions.

**Funding** This work was supported through the Developing Excellence in Leadership, Training and Science (DELTAS) Africa Initiative grant # DEL-15-011 to THRiVE-2. The DELTAS Africa Initiative is an independent funding scheme of the African Academy of Sciences (AAS)'s Alliance for Accelerating Excellence in Science in Africa (AESA) and supported by the New Partnership for Africa's Development Planning and Coordinating Agency (NEPAD Agency) with funding from the Wellcome Trust grant #107742/Z/15/Z and the UK government.

**Disclaimer** The views expressed in this publication are those of the author(s) and not necessarily those of AAS, NEPAD Agency, Wellcome Trust or the UK government.

**Competing interests** None declared.

**Patient consent for publication** Not required.

**Ethics approval** We obtained permission to use the data sets from the DHS programme. During the survey data collection process, implemented by the Uganda Bureau of Statistics, written informed consent was obtained from all the respondents.

**Provenance and peer review** Not commissioned; externally peer reviewed.

**Data availability statement** Data are available in a public, open access repository. All data relevant to the study are included in the article or uploaded as supplemental information. Data used for this analysis can be accessed, with permission, from the Demographic and Health Survey programme website https://www.dhsprogram.com/data/available-datasets.cfm.

**ORCID iD**
Dinah Amongin http://orcid.org/0000-0002-1420-005X

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
