## [Reviewer comments · BMJ Open]

ARTICLE DETAILS

TITLE (PROVISIONAL)	Later Life Outcomes of Women by Adolescent Birth History: Analysis of the 2016 Uganda Demographic and Health Survey
AUTHORS	Amongin, Dinah; Kågesten, Anna; Tunçalp, Özge; Nakimuli, A; Nakafeero, Mary; Atuyambe, Lynn; Hanson, Claudia; Benova, Lenka

VERSION 1 – REVIEW

REVIEWER	Yohannes Wado African Population and Health Research Center
REVIEW RETURNED	25-Jul-2020

GENERAL COMMENTS	This paper looks at an important reproductive health issue for countries like Uganda with a high fertility rate and higher adolescent fertility in particular. The findings underscore the effect of early birth and repeat adolescent births on later life reproductive and socio-economic outcomes. The manuscript is well-written. One of my concerns relates to the number of outcomes (reproductive, socio-economic, and many others) the authors looked at in this particular paper, and little or no theoretical background is provided for many of these outcomes. I wonder how early childbearing and closely spaced births before age 20 relate to Condom use at last sex, being head of household, occupation, self-reported STI and HIV testing for women currently aged 40-49. A brief literature review on the relationship between some of these outcomes and the variables of interest is needed. In other words, there should be a rationale or a systematic approach to the choice of these outcome variables. I also suggest focusing on a few reproductive and socio-economic outcome variables. In addition, much of the results presented in the manuscript come from a test of proportions and p-values. Will these associations with early and repeat childbearing hold in a multivariate analysis that adjusts for other demographic and socio-economic characteristics (maybe I missed the regression tables)?
---

REVIEWER	Lynn Matthews University of Alabama at Birmingham USA
REVIEW RETURNED	20-Aug-2020

GENERAL COMMENTS	Interesting manuscript looking at unadjusted relationships using cross-sectional 2016 Uganda DHS data collected from women in their 4th generation of life to evaluate relationship between their report of pregnancy before 18 years of and SES/education/SRH now (in their 40s). Predictors of interest are self report of having a child before 18 years of age and outcomes of interest are current
---

	socioeconomic status, education, and lifetime reproductive outcomes. The nature of the data do not allow for appropriate adjustment for the many confounders of having a child as an adolescent and SES and educational and measures of SRH 20+ years later, but the associations are of interest. There is an emphasis on stratification by marital status at the time of the earlier pregnancy as well as the current marital status that is not well justified. Would be improved by growing discussion and clarifying/honing results as per below specific comments. Abstract:  - Consider rewording the objectives which are not clear from the abstract. For example, “initiation <18 and repeat <20 years” makes sense after reading the paper but not as the first sentence of the abstract. - “Outcome measures” section defines predictors of interest but not outcomes. - “Conclusions” – this design does not allow for causal inference - would reword the conclusions. Introduction:  - Please expand on the sentence on page 5 lines 106-108 that summarizes the prior data on this topic in order to share a bit of what has been done and what is known. This analysis seems tightly linked to ref 29 so would be helpful to share what is already known and how this work grows/expands the knowledge base. Methods:  - Childbearing (having a child) is not the same as fertility (conceiving a child.) Would review carefully to make sure language is appropriate. Title mentions fertility pattern but predictors for this analysis are reports of having had a child. - Overall, tables are a bit overwhelming. Would consider honing to key variables and avoid repetition (E.g. completed schooling categorical, mean years of schooling, literacy are all related maybe pick most important or at least drop one. Similarly for mean # live births, mean # live births limited to 20-40. Unmet need and current use of contraception highly related so would pick one. Etc.) This also relates to comment below about justifying the outcomes evaluated. - The authors discuss possibility of recall bias in the reporting of adolescent pregnancies. A bigger limitation (or additional important limitation) may include social desirability bias as women are likely to remember having a child, but may be less likely to report it if pregnancy in adolescence is stigmatized for the participants. - Justify/explain why these data were stratified once by marital status at the time of the first pregnancy and then again in another analysis by current marital status during the 2016 survey. - Page 7 lines 150-158: Justify why these variables chosen. Some are not intuitive (condom use? BMI?) and would be helpful to know what model or thinking or prior work informed these choices of outcomes. - There are a few different representations of household wealth index – Supp Table 2 mentions quintiles for calculating household wealth, Table 5 reports a median score, - While this team did not collect the DHS data, information on how it is collected / how sampling is designed for Uganda would help to interpret the strengths and limitations of the dataset used to conduct the analysis. Results:  - Consider displaying the frequency data (currently in Table 1) with a
--	--

	chart. Lack of figures/charts make this harder to digest.  - Worth noting somewhere the mean age at first birth which is 19 for the entire group and closer to 15/16 for those with first birth <18. - Line 226, “mean age at ... first birth...differed between groups.” But the groups are women with different first birth.? Check. - page 12, line 269 – I think the point is that women with later birth had higher proportion of partners with secondary education. But sentence says something else. Check. - page 12 line 270, say which group had higher/lower/more frequent justification of wife beating Discussion:  - It would help to provide some context about what the sociocultural context is for having children at a young age in Uganda. What happens if the woman does not marry the man who fathered the child? Is the child raised by the woman/girl and her birth family? Is the child raised by the family of the man believed to have fathered the child? Is the expectation that once she has a child with a man, she becomes his family’s financial responsibility? Is she expected to move to his family’s land/home even if far? How might this have been handled in the 90s when the referent children were born? - Can the authors comment on strategies for achieving some of the suggested policy changes e.g. “adolescent women should be protected from early marriage” and “improved access to continued schooling, higher education, and effective contraceptive services” are needed to support girls with a child born before she reaches 18 years of age. (Especially since MOH involved in this work.) - A discussion of barriers and promoters of repro health autonomy (e.g. contraception, sexual debut, abortion access) may be appropriate since the conclusion is to try and prevent childbearing among young women in Uganda. - The term school “drop-out” implies that the student chose to not complete school but I suspect for many of these girls/women the issue was less her choice and more about who would fund school, who would watch child, etc. Consider re-wording. - Would review the discussion keeping in mind the point authors made in limitaitons about how causality cannot be inferred in this paper and being aware of how much confounding is not accounted here between age at first child and socioeconomic status in 4th decade of life. There are some very strong conclusions based on very scant associations. Minor editorial points:  - Consider Uganda DHS instead of UDHS. Only one extra word and easier on the reader throughout.
--	--

VERSION 1 – AUTHOR RESPONSE

Reviewer 1:

This paper looks at an important reproductive health issue for countries like Uganda with a high fertility rate and higher adolescent fertility in particular. The findings underscore the effect of early birth and repeat adolescent births on later life reproductive and socio-economic outcomes. The manuscript is well-written. Thank you so much for this feedback.

NA

One of my concerns relates to the number of outcomes (reproductive, socio-economic, and many

others) the authors looked at in this particular paper, and little or no theoretical background is provided for many of these outcomes. I wonder how early childbearing and closely spaced births before age 20 relate to Condom use at last sex, being head of household, occupation, self-reported STI and HIV testing for women currently aged 40-49. A brief literature review on the relationship between some of these outcomes and the variables of interest is needed. In other words, there should be a rationale or a systematic approach to the choice of these outcome variables. I also suggest focusing on a few reproductive and socio-economic outcome variables. Thank you very much for this important comment. Following your guidance, we have reviewed literature and have added more information on paragraph 4, page 5&6, which explain how early pregnancy impacts on later life of a woman.

Further, we have revised the variables and have focused on a few key reproductive health and socioeconomic outcomes. Please see adjustments in the methods section, Results and tables & supplementary file 2. Supplementary table 3 is now deleted. Lines 107-121, page 5 & 6

Methods, results sections.

In addition, much of the results presented in the manuscript come from a test of proportions and p-values. Will these associations with early and repeat childbearing hold in a multivariate analysis that adjusts for other demographic and socio-economic characteristics (maybe I missed the regression tables)?

Thank you for this observation. You did not miss the regression tables.

We did not do this because we did not want to 1) imply causality – the nature of the dataset does not allow us to consider the causal nature of associations, and 2) we did not have one concrete outcome in mind on which to build a well-adjusted model. NA

REVIEWER 2:

Interesting manuscript looking at unadjusted relationships using cross-sectional 2016 Uganda DHS data collected from women in their 4th generation of life to evaluate relationship between their report of pregnancy before 18 years of age and SES/education/SRH now (in their 40s). Predictors of interest are self report of having a child before 18 years of age and outcomes of interest are current socioeconomic status, education, and lifetime reproductive outcomes. The nature of the data do not allow for appropriate adjustment for the many confounders of having a child as an adolescent and SES and educational and measures of SRH 20+ years later, but the associations are of interest.

Thank you very much. NA

There is an emphasis on stratification by marital status at the time of the earlier pregnancy as well as the current marital status that is not well justified. Thank you very much for noting this. We have added more literature on impact of early marriage on long term outcomes of women (lines 95-97, page 5). Among women with first birth <18 years (with and without repeat adolescent birth), we stratified analysis by marital status at time of first birth (lines 129-132, page 6).

Regarding analysis among women married at the point of survey, we have corrected the use of the term “stratification” as we did not actually stratify. We conducted additional analysis among this subsample as it contained partner variables that are lacking in the entire sample of women. We have provided literature in the introduction that justifies our interest in the current partner variables. In the literature, women with early birth tend to end up with partners of lower socioeconomic achievements

(lines 117-119).
Lines 95-97, page 5

Lines 129-132, page 6

Lines 165-170 & 185-188, 250-252, 261-262
pages 8, 9, 12

Lines 117-119, page 6

Would be improved by growing discussion and clarifying/honing results as per below specific comments.

Thank you for this detailed feedback. We have addressed each comment below. NA

Abstract

- Consider rewording the objectives which are not clear from the abstract. For example, "initiation <18 and repeat <20 years" makes sense after reading the paper but not as the first sentence of the abstract.

- "Outcome measures" section defines predictors of interest but not outcomes.

- "Conclusions" – this design does not allow for causal inference - would reword the conclusions.

Thank you so much for this guidance. This have done this- objective, measures and conclusion.

Lines 25-26, 30, 46-47

Pages 2 &3

Introduction

- Please expand on the sentence on page 5 lines 106-108 that summarizes the prior data on this topic in order to share a bit of what has been done and what is known. This analysis seems tightly linked to ref 29 so would be helpful to share what is already known and how this work grows/expands the knowledge base.

This is well appreciated. We have provided more information on what was done and is known about the topic. Lines 107-121

Pages 5&6,

Methods

- Childbearing (having a child) is not the same as fertility (conceiving a child.) Would review carefully to make sure language is appropriate. Title mentions fertility pattern but predictors for this analysis are reports of having had a child.

Thank you for this important clarification. We have proceeded and used 'birth' instead of fertility.

'Adolescent fertility pattern' replaced with 'adolescent birth history'. We made the necessary changes throughout the document, right from the title.

NA

- Overall, tables are a bit overwhelming. Would consider honing to key variables and avoid repetition (E.g. completed schooling categorical, mean years of schooling, literacy are all related maybe pick most important or at least drop one. Similarly for mean # live births, mean # live births limited to 20-40. Unmet need and current use of contraception highly related so would pick one. Etc.) This also relates to comment below about justifying the outcomes evaluated.

This guidance is much appreciated. We have deleted the repeated variables. Please see the revised Methods and Results sections. Supplementary file 3 has also been deleted. Methods and results sections.

- The authors discuss possibility of recall bias in the reporting of adolescent pregnancies. A bigger limitation (or additional important limitation) may include social desirability bias as women are likely to remember having a child, but may be less likely to report it if pregnancy in adolescence is stigmatized for the participants.

Thank you. To assess any differential reporting of adolescent childbearing, we conducted a sensitivity analysis to examine recall of adolescent birth history of women born between 1967-1976 in the 1995 vs 2016 survey (lines 149-154), it was found to be similar and therefore pointing to minimal differential reporting of adolescent births on a population level (the full findings are provided in Supplementary file 1).

Following your comment, we have included more information and provided a citation regarding age heaping in DHS reports of age at first birth. Lines 149-154, page 7

Line 333-336, page 15

- Justify/explain why these data were stratified once by marital status at the time of the first pregnancy and then again in another analysis by current marital status during the 2016 survey.

Thank you very much for noting this. We have added more literature on impact of early marriage on long-term outcomes of women (lines 95-97, page 5). Among women with first birth <18 years (with and without repeat adolescent birth), we stratified analysis by marital status at time of first birth (lines 129-132, page 6)..

Regarding analysis among women married at the point of survey, we have corrected the use of the term "stratification" as we did not actually stratify. We conducted additional analysis among this subsample as it contained partner variables that are lacking in the entire sample of women. We have provided literature in the introduction that justifies our interest in the current partner variables. In the literature, women with early birth tend to end up with partners of lower socioeconomic achievements (lines 117-119).

Lines 95-97, page 5

Lines 129-132, page 6

Lines 165-170 & 185-188, 250-252, 261-262
pages 8, 9, 12

Lines 117-119, page 6

- Page 7 lines 150-158: Justify why these variables chosen. Some are not intuitive (condom use? BMI?) and would be helpful to know what model or thinking or prior work informed these choices of outcomes.

This guidance is much appreciated. Based on this, we have reviewed the literature again and provided more information for the choice of variables (now a reduced set of key outcomes). Lines 105-121, page 5&6

- There are a few different representations of household wealth index – Supp Table 2 mentions quintiles for calculating household wealth, Table 5 reports a median score, In the DHS, both the household wealth score (a continuous variable) and household wealth quintiles information are available. We have opted to retain wealth score. This has been updated in the Methods and Results sections.

Line 160, page 6

Supplementary file 2

- While this team did not collect the DHS data, information on how it is collected / how sampling is designed for Uganda would help to interpret the strengths and limitations of the dataset used to conduct the analysis.

Thank you for this important observation. We have included this information at the start of the Methods section. Lines 136-142, page 7

Results:

- Consider displaying the frequency data (currently in Table 1) with a chart. Lack of figures/charts make this harder to digest.

We appreciate this advice. Accordingly, we have presented this information in a chart- Fig 1. Page 21

- Worth noting somewhere the mean age at first birth which is 19 for the entire group and closer to 15/16 for those with first birth <18.

Thank you. We have added this. Lines 228-229, page 11

- Line 226, “mean age at ... first birth...differed between groups.” But the groups are women with different first birth.? Check.

We have clarified on this. Thank you so much for noting it. Line 221-225, pages 10/11

- page 12, line 269 – I think the point is that women with later birth had higher proportion of partners with secondary education. But sentence says something else. Check.

Thank you for noting this error. We have corrected it. Lines 252-254, page 12

- page 12 line 270, say which group had higher/lower/more frequent justification of wife beating

Thank you for noting this. Due to a review on the variables to be included, we opted to delete this outcome. Table 2

Discussion

- It would help to provide some context about what the sociocultural context is for having children at a young age in Uganda. What happens if the woman does not marry the man who fathered the child? Is the child raised by the woman/girl and her birth family? Is the child raised by the family of the man believed to have fathered the child? Is the expectation that once she has a child with a man, she becomes his family’s financial responsibility? Is she expected to move to his family’s land/home even if far? How might this have been handled in the 90s when the referent children were born?

Thank you for this important guidance. We have provided information on the context in the conclusion.

Lines 369-378, page 17

- Can the authors comment on strategies for achieving some of the suggested policy changes e.g. “adolescent women should be protected from early marriage” and “improved access to continued schooling, higher education, and effective contraceptive services” are needed to support girls with a child born before she reaches 18 years of age. (Especially since MOH involved in this work.)

This guidance is much appreciated. We have provided some specifics on the strategies for achieving the suggested policy changes. Lines 355-360, Page 16&17

- A discussion of barriers and promoters of repro health autonomy (e.g. contraception, sexual debut, abortion access) may be appropriate since the conclusion is to try and prevent childbearing among young women in Uganda.

We have added this information. Please find in the discussion section. Lines 320-328, page 15.

- The term school “drop-out” implies that the student chose to not complete school but I suspect for many of these girls/women the issue was less her choice and more about who would fund school, who would watch child, etc. Consider re-wording. We appreciate this guidance. We have reworded this to ‘school discontinuation’. Line 288, page 13

- Would review the discussion keeping in mind the point authors made in limitaitons about how causality cannot be inferred in this paper and being aware of how much confounding is not accounted here between age at first child and socioeconomic status in 4th decade of life. There are some very strong conclusions based on very scant associations.

Thank you so much for this important comment. We have adjusted the wording in the conclusion to reflect this limitation.

Line 344-351, page 16

Minor editorial points:

- Consider Uganda DHS instead of UDHS. Only one extra word and easier on the reader throughout.

This guidance is much appreciated. We have adopted this. NA